# Trends in cost and consumption of essential medicines for non-communicable diseases in Azerbaijan, Georgia, and Uzbekistan, from 2019 to 2021

Ninell Kadyrova[1], Dzintars Gotham[1], Stanislav Kniazkov[1]*, Elsever Aghayev[2,3], Polad Hajibalayev[3], Zohid Ermatov[4], Natasha Azzopardi Muscat[5]

1 Pricing and Reimbursement Analytics Team, Regional Office for Europe, World Health Organization, Copenhagen, Denmark, 2 Pharmacological and Pharmacopoeial Expert Council, Ministry of Health of the Republic of Azerbaijan, Baku, Azerbaijan, 3 Pharmaceutical Faculty, Azerbaijan State Advanced Training Institute for Doctors named after A. Aliyev, Baku, Azerbaijan, 4 State Medical Insurance Fund, Tashkent, Uzbekistan, 5 Health Policies and Systems, Regional Office for Europe, World Health Organization, Copenhagen, Denmark

* stanislav.kniazkov@gmail.com

**Data Availability Statement:** Data can be found in the Dryad repository, at doi.org/10.5061/dryad.7wm37pw05.

## Abstract

### Background

Access to medicines is a global priority. Azerbaijan, Georgia, and Uzbekistan have different approaches to pricing policies for pharmaceuticals. The aim of this study was to analyze recent trends in the consumption and prices of non-communicable disease (NCD) medicines in Azerbaijan, Georgia, and Uzbekistan, in the outpatient setting.

### Methods

We included medicines for asthma and COPD, cancer, cardiovascular disease, diabetes, epilepsy, and mental disorders. Sales data for pharmaceutical products in community pharmacies were extracted from a commercial database. Changes in consumption and prices were analyzed across all included NCD medicines, by disease category and pharmacological group.

### Results

Consumption of NCD medicines was highest in Georgia, at twice the levels in Azerbaijan, and four times levels in Uzbekistan. Average prices of NCD medicines, weighted by consumption, increased by 26% in Georgia, but decreased by 3% in Azerbaijan and by 0.1% in Uzbekistan. Prices increased for all disease groups in Georgia (from +13% for epilepsy medicines to +86% for cancer), varied by group in Uzbekistan (from -22% for epilepsy medicines to +47% for cancer), while changes in Azerbaijan were smaller in magnitude (from -4% for medicines for cardiovascular disease to +11% for cancer). Cancer medicines had markedly higher prices in Uzbekistan, and asthma and COPD medicines had markedly higher prices in Azerbaijan and Uzbekistan.

**Funding:** This study was funded by WHO Europe.

**Competing interests:** NK has received funding from the World Health Organization, Health Transform Forum/GIZ, Childhood without diabetes/World Diabetes Foundation and declares no competing interests. DG has received funding from the World Health Organization, Global Justice Now, Medicines Patent Pool, Médecins Sans Frontières, STOPAIDS, Treatment Action Group, and the World Intellectual Property Organization, and declares no competing interests. SK is a salaried employee of WHO Europe and declares no conflict of interest. EA receives funding from the Azerbaijan State Advanced Training Institute for Doctors named after A.Aliyev and declares no conflicts of interest. PH is a PhD researcher of Pharmaceutical Faculty of the Azerbaijan State Advanced Training Institute for Doctors named after A. A. Aliyeva and declares no conflicts of interest. ZE receives money from the State Medical Insurance Fund of Uzbekistan in the form of a salary and declares no conflicts of interest. NAM is a salaried employee of WHO Europe and declares no conflict of interest.

**Abbreviations:** NCD, non-communicable disease; COPD, chronic obstructive pulmonary disease; EECA, Eastern Europe and Central Asia; HIV, human immunodeficiency virus; WHO, World Health Organization; OOP, out-of-pocket; TB, tuberculosis; SMD, Support in Market Development; DDD, defined daily dose; DID, number of defined daily doses consumed per 1,000 inhabitants per day; OECD, Organization for Economic Co-operation and Development; DALY, disability-adjusted life year; Q, quarter; GDP, gross domestic product; EPR, external price referencing; UHCP, universal health care programme.

## Conclusions

Georgia showed the highest outpatient consumption of NCD medicines, suggesting the broadest access to treatment. However, Georgia also saw marked price increases, greater than in the other countries. In Georgia, where there was no price regulation, widespread price increases and increases in consumption both contribute to increasing pharmaceutical expenditures. In Azerbaijan and Uzbekistan, increases in outpatient pharmaceutical expenditures were primarily driven by increases in consumption, rather than increases in price. Comparing trends in consumption and pricing can identify gaps in access and inform future policy approaches.

## Introduction

Increasing access to affordable medicines is a priority for health systems globally and forms a key part of delivering universal health coverage. Especially in the case of medicines on the WHO Essential Medicines List, regular and affordable access is crucially important for delivering good care to those living with chronic medical conditions.

This study analyses trends in the consumption and prices of medicines for NCDs in three countries in the Eastern Europe and Central Asia (EECA) region: Azerbaijan, Georgia, and Uzbekistan. NCDs were responsible for 71% of deaths in low- and middle-income countries in 2019 [1] and scaling up the management of NCDs is a key priority in the EECA region [2].

All three countries have national health systems, which are predominantly funded through taxes, although medicines are mainly paid for out-of-pocket. In Azerbaijan, state health insurance provides free services at state health facilities, although many specialist services are not covered. Certain inpatient medicines are provided for free, and outpatient medicines for certain state health programs (e.g. for cancer, diabetes, and TB) are provided for free, with other inpatient and outpatient medicines covered out of pocket [3].

In Uzbekistan, the state health insurance package includes primary care, emergency care, and specialized care for vulnerable groups, as well as care for certain "socially significant and hazardous conditions", including diseases such as HIV and certain noncommunicable diseases such as cancer. WHO reports that the "package largely excludes secondary and tertiary care, as well as outpatient pharmaceuticals, for significant parts of the population" [4].

In Georgia, coverage is nearly universal, but covers only a small package of health products and services [5].

At the same time, out-of-pocket spending as a proportion of total health expenditures on health is high, standing at 79% in Azerbaijan, 56% in Georgia, and 52% in Uzbekistan [5]. Considering expenditures on medicines specifically, 56% of household health expenditures are on pharmaceuticals in Azerbaijan (2015 data) [6], 65% in Georgia (2018 data) [5] and up to 38% in Uzbekistan (proportion represents spending on medical goods, 2018) [4].

High out-of-pocket expenditures on pharmaceuticals represent a barrier to accessing healthcare and can create or exacerbate financial hardships [7]. Rising medicines prices create ever increasing pressure on household budgets, particularly when a household member requires long-term treatment, as many people living with non-communicable diseases (NCDs) will. Expenditure on pharmaceuticals represents a significant portion of national healthcare expenses, particularly in low- and middle-income countries, where out- of-pocket (OOP) expenditure made up 35% of health expenditures in 2019 [8].

Azerbaijan has 'full' price regulation for all registered medicines, wherein the national Tariff (Price) Council sets prices at various points along the supply chain (including regulation of ex-factory, wholesale, and pharmacy retail prices), employing both external price referencing and internal price referencing. Inpatient medicines are reimbursed, and outpatient medicines are reimbursed for certain defined diseases [5]. In 2019–2021, Georgia had no form of price regulation, a policy approach sometimes described as 'free pricing'. The costs of inpatient medicines are covered by the national health system as part of coverage for hospital treatment. Outpatient medicines are reimbursed in some cases, based on the disease, social status, and age of the patient [5]. Until 2018, Uzbekistan used 'full' price regulation along the supply chain, but only for 112 "socially important" medicines, covering a number of NCDs–cancer, diabetes, and mental health disorders–as well as TB, HIV, leprosy, and sexually transmitted diseases [5]. This mechanism was overhauled in 2018–2020 and, since 2020, Uzbekistan sets price limits for all medicines, using external price referencing [9]. Additionally, Uzbekistan regulates wholesaler and pharmacy markups for all medicines, with limits of 15% and 20% markups, respectively. Outpatient medicines are covered if they are on the list of 'socially important' medicines for 13 defined diseases [5]. A pilot of reimbursement of 11 medicines for arterial hypertension, diabetes and asthma has started in the Syrdaria oblast of Uzbekistan.

Prevention and treatment services for NCDs were severely disrupted since the COVID-19 pandemic began, and many people had no access to treatment for hypertension, heart attacks, strokes, cancer, or diabetes. Fifty-three percent of countries reported disruption in services for hypertension treatment, 49% for services for diabetes and diabetes-related complications, 42% for cancer treatment, and 31% for cardiovascular emergencies [10]. During the pandemic, many families experienced a fall in their incomes and this also impacted their ability to required medicinal products in countries with poorly or insufficiently developed drug pricing regulation systems [11]. The COVID-19 crisis has added to the growing understanding that the scarcity of many essential medicines is not inevitable but rather the consequences of ineffective policies [12].

The aim of this study was to analyze recent trends in the consumption and prices of NCD medicines in Azerbaijan, Georgia, and Uzbekistan, in the outpatient setting. Understanding these trends is an integral part of improving the accessibility and affordability of medicines for NCDs, enabling better management, and improving health outcomes.

## Materials and methods

Data on medicines sales volumes and prices were sourced from the Support in Market Development (SMD) database [13]. The database provides data for sales volumes and prices for medicines dispensed from community pharmacies, by quarter.

The list of included NCD medicines was based on the WHO package of essential noncommunicable disease interventions for primary health care (WHO PEN). Given different market availability for different medicines, the final number of medicines identified in each country were 181 medicines in Azerbaijan, 185 in Georgia and 153 in Uzbekistan (Table 1). These include oral diabetes medicines, cardiovascular medicines, medicines for asthma and chronic obstructive pulmonary disease (including antibiotics that are commonly used in their management), cancer medicines (predominantly hormone therapies for breast and prostate cancer), epilepsy medicines, and medicines for mental disorders (predominantly antidepressants and anxiolytics). For Azerbaijan and Georgia, three years of data (2019–2021) were available, while for Uzbekistan, only two years (2019–2020) were available.

Prices were standardized to US dollars per defined daily dose (DDD). One DDD represents a typical daily amount used by a patient, based on standards published by the World Health

**Table 1. Number of NCD medicines with available data on consumption and prices, by disease category and country.**

| Disease category | Azerbaijan | Georgia | Uzbekistan |
|---|---|---|---|
| Diabetes | 15 | 14 | 14 |
| Cardiovascular disease | 91 | 90 | 72 |
| Asthma and COPD, including relevant antibiotics | 37 | 35 | 32 |
| Cancer | 8 | 13 | 10 |
| Epilepsy | 10 | 10 | 9 |
| Mental disorders | 20 | 23 | 16 |
| Total | 181 | 185 | 153 |

Organization. Using cost per DDD enables comparison of average costs between disease groups [14]. Trends in consumption were expressed using a metric that is standardized to population size: number of DDD consumed per 1,000 inhabitants per day (DID).

Average prices presented in this analysis are weighted means unless otherwise specified, to avoid distortion of means by high-priced medicines that are only consumed at low volumes. Changes in consumption and prices over 2019–21 were analyzed in terms of absolute and relative changes over time, across all included NCD medicines as well as by disease category and by pharmacological group.

## Results

### Consumption of NCD medicines

Overall consumption of NCD medicines increased by 27% in Azerbaijan (2019–21), by 16% in Georgia (2019–21), and by 25% in Uzbekistan (2019–20) (Fig 1). The highest consumption was seen in Georgia, and consumption in Georgia was higher than in the other two countries across all NCD therapeutic groups, except for asthma and COPD medicines (Figs 1 and 2).

Consumption of antihypertensives in Georgia (188 DID in 2021) was double that in Azerbaijan and five times the level in Uzbekistan (Fig 2), similar to levels in Türkiye (154 DID in 2019) and Latvia (203 DID in 2020), but lower than the OECD-29 average (328 DID in 2019)

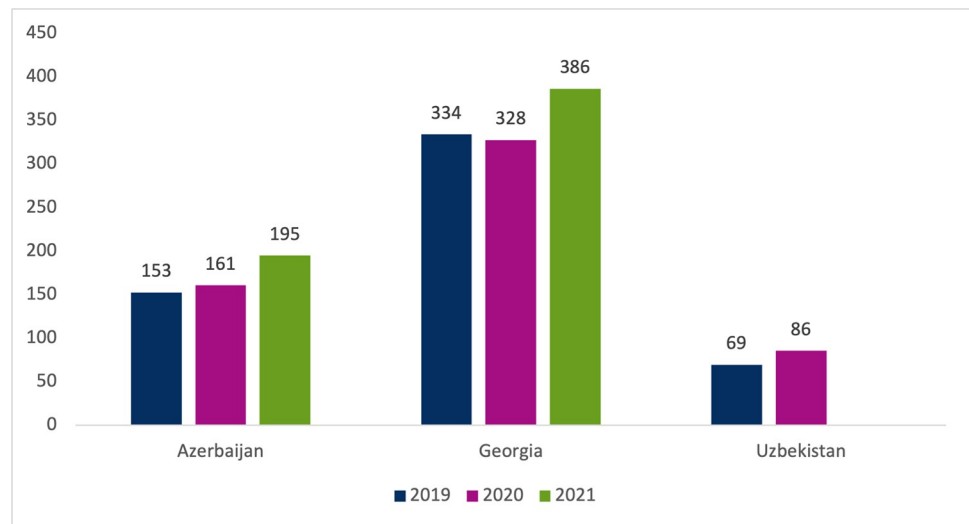

**Fig 1. Consumption of NCD medicines in Azerbaijan, Georgia, and Uzbekistan in 2019–2021, outpatient setting, DDDs per 1000 inhabitants per day.** DDD–defined daily dose.

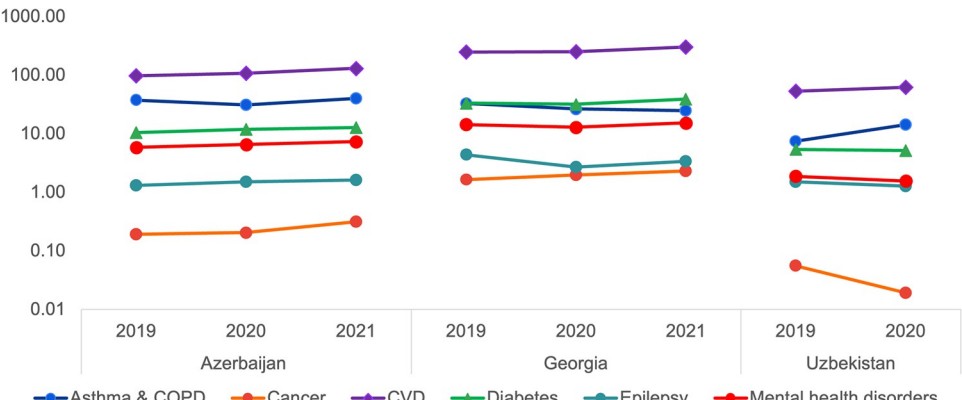

**Fig 2. Consumption of NCD medicines, DDD per 1000 inhabitants per day, by specific health conditions in 2019–2021.**

[15]. Consumption of oral diabetes medicines in Georgia was 10 times higher than in Uzbekistan in 2020 (54 versus 5 DID) and 4 times higher than in Azerbaijan (12 DID) in 2021. Consumption of hormone treatments for cancer was 7 times higher in Georgia than in Azerbaijan (2.3 versus 0.3 DID) and over 120 times higher than in Uzbekistan, in 2020. Consumption of medicines for anxiety disorders in Georgia was 6.73 DID–over 8 times higher than in Azerbaijan (0.77 DID) and 13 times higher than in Uzbekistan (0.49 DID). While antidepressant consumption in Georgia (8.6 DID) was 24% higher than in Azerbaijan and 8 times higher than in Uzbekistan, it was 7.7 times lower than in OECD countries (66 DID) in 2019 and 2.7 times lower than in Latvia (18 DID) [15]. At the same time, consumption of bronchodilators in Azerbaijan was 2.3 times higher than in Georgia in 2021 and 6.8 times higher than in Uzbekistan in 2020 (Fig 2 and S2 File).

Analysis of quarterly trends in the consumption of NCD medicines revealed decreases in consumption in the second quarter of 2020: reductions of 27% in Azerbaijan, 11% in Georgia, and 7% in Uzbekistan. These dips in consumption coincide with the beginning of the COVID-19 pandemic in early 2020 [16]. It is worth noting that the most pronounced 2020 Q2 decreases were seen for asthma and COPD medicines (decreases of 72% in Azerbaijan, 39% in Georgia, 47% lower in Uzbekistan). In all three countries, these dips were followed by a return to baseline consumption in Q3 and Q4 of 2020.

Precipitous dips in consumption were observed already before the COVID-19 pandemic, for some medicine groups in some quarters. The frequency of dips in consumption (proportion of medicines in each disease category with a decrease in consumption of more than 25% quarter-on-quarter) are shown in Fig 3, by country, disease, and quarter. Overall, quarter-on-quarter dips in consumption were seen most frequently in Uzbekistan, with around dips seen for 20–50% of medicines in each therapeutic category, depending on the quarter. Comparing different therapeutic categories, dips in consumption were seen most frequently in medicines for asthma and COPD in Azerbaijan and Georgia, while in Uzbekistan, the frequency of dips in consumption was more evenly distributed among disease categories.

In terms of average reduction in consumption, greatest quarter-on-quarter changes in consumption were seen in Uzbekistan in 2019 Q4 –with reductions of 59% for hormone treatments for cancer, 52% for mental disorder medicines, 39% for cardiovascular disease (CVD) medicines, and 38% for diabetes treatments (S2 File). As these changes preceded the COVID-19 pandemic, they likely represent interruptions in supply due to country-specific factors.

### Azerbaijan

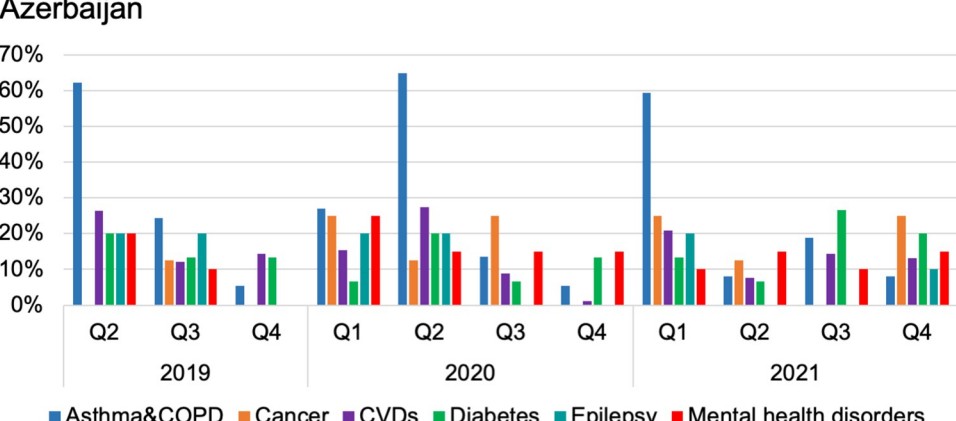

### Georgia

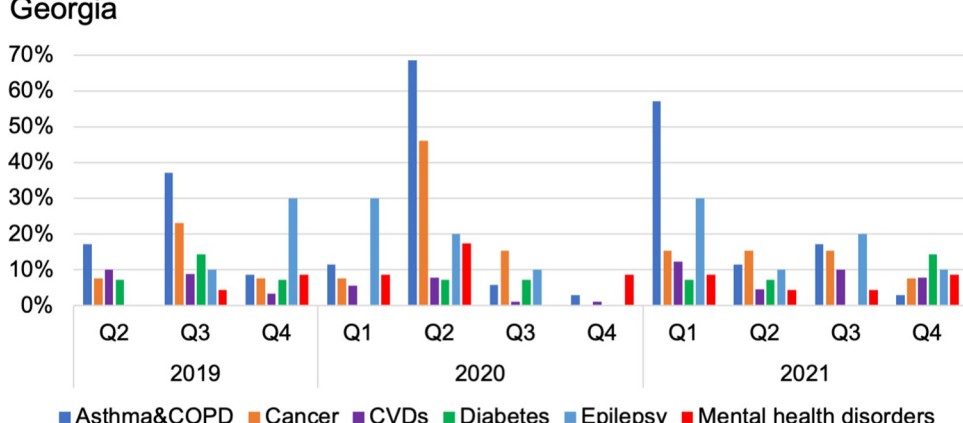

### Uzbekistan

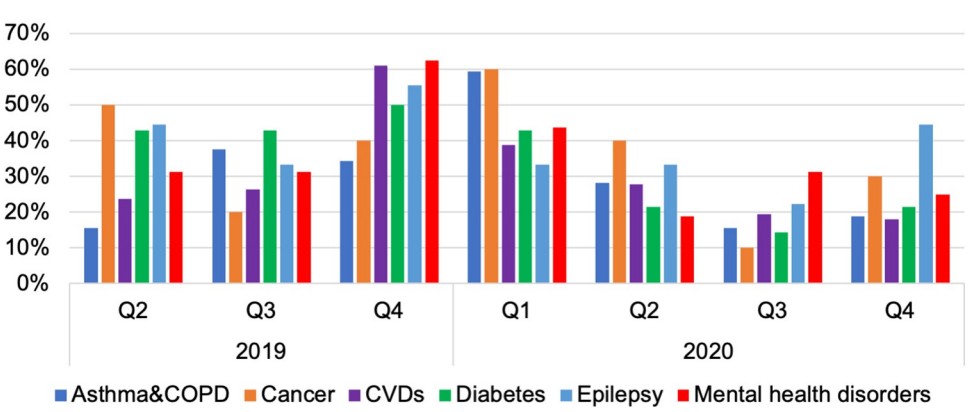

**Fig 3. Proportion of medicines with significant dips in consumption, by disease and quarter.**

## Trends in average cost per day of treatment

For CVD, diabetes, epilepsy, and mental disorder medicines, average costs per DDD were similar across the three countries (Fig 4). For asthma and COPD medicines, costs were markedly lower in Azerbaijan and higher in Georgia and Uzbekistan: for the most recent year with data available, the price per 1 DDD was 81% higher in Uzbekistan than in Azerbaijan, and 205%

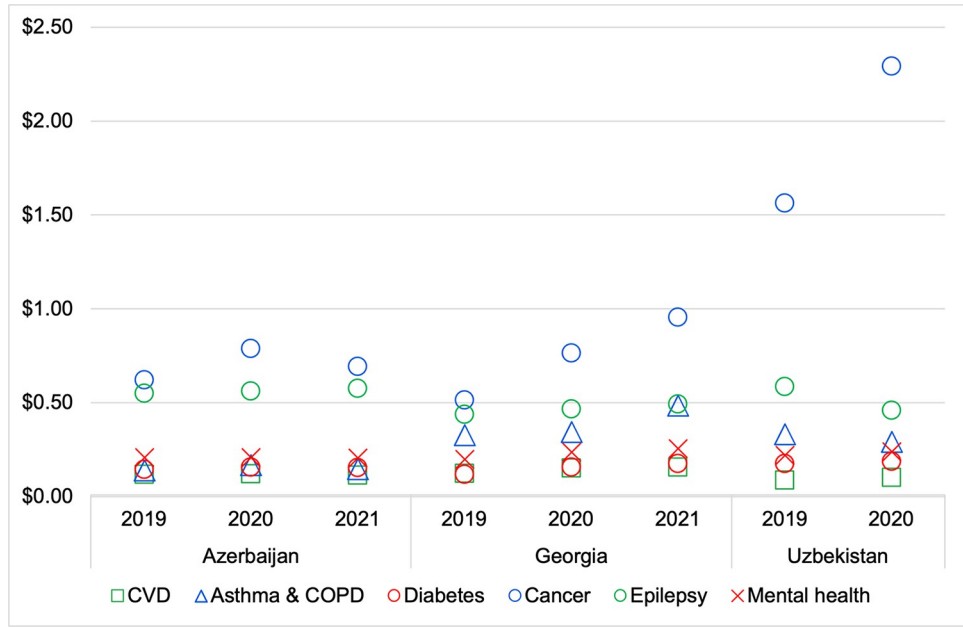

**Fig 4. Average cost of 1 day of treatment with NCD medicines in 2019–2021, USD.** Costs shown are average cost per DDD weighted by the volume of consumption for each medicine included in the category.

higher in Georgia than in Azerbaijan. For cancer medicines, costs were similar in Azerbaijan and Georgia but markedly higher in Uzbekistan, where the prices in the most recent year with data available were 140% above prices in Georgia and 231% above prices in Azerbaijan.

The average prices of NCD medicines, weighted by consumption, increased by 26% in Georgia, but decreased by 3% in Azerbaijan and by 0.1% in Uzbekistan (Fig 5).

Changes within individual therapeutic groups were more pronounced (Fig 6). Azerbaijan, prices increased for medicines for asthma and COPD (+7%), hormone therapies for cancer (+11%), oral diabetes medicines (+8%), and epilepsy (+5%), while decreasing for CVD medicines (-4%) and anxiety and depression (-1%). Prices increased in all therapeutic groups in Georgia, ranging from an increase of 13% in the average price of epilepsy medicines, to an 86% increase for hormone therapies for cancer. In Uzbekistan, prices increased for hormone therapies for cancer (+47%), CVD medicines (+12%), oral diabetes medicines (+6%), and medicines for anxiety and depression (+8%), while decreasing for asthma and COPD (-12%) and epilepsy (-22%).

Overall expenditure on the NCD medicines included in this analysis increased from US$72 to 89 million in Azerbaijan (+24%), from US$61 to 89 million in Georgia (+45%), and from US$100 to 119 million in Uzbekistan (+18%).

## Medicines with the highest overall expenditures

In all three countries, the top 20 medicines by overall expenditure represented over half of all expenditures captured in this analysis (Table 2). In Georgia, the top 20 (of a total 185 medicines with data available) made up 59% of total expenditures. In Azerbaijan, the top 20 (of a total 181 medicines with data available) made up 61% of total expenditures. In Uzbekistan, the top 20 (of a total 153 medicines with data available) made up 64% of total expenditures. Among the 20 medicines with the greatest overall expenditure, 15 were CVD medicines in Georgia, 11 in Azerbaijan, and 10 in Uzbekistan (Table 2).

## Azerbaijan

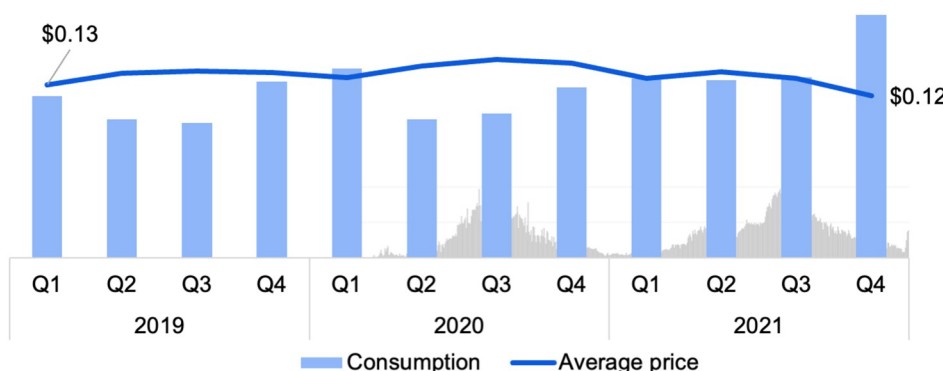

## Georgia

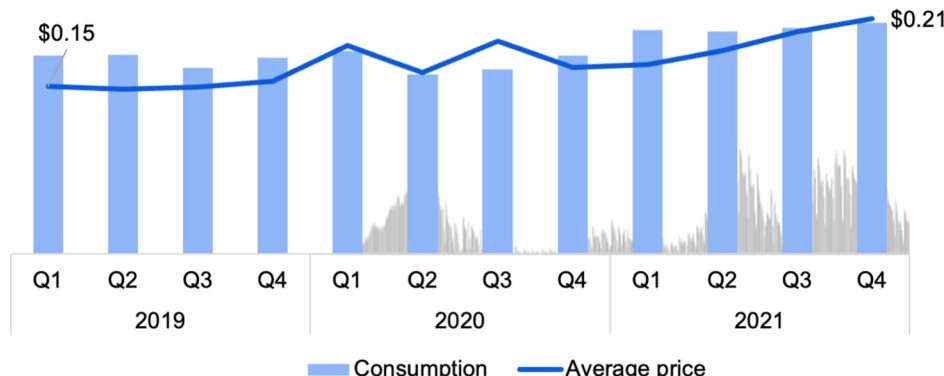

## Uzbekistan

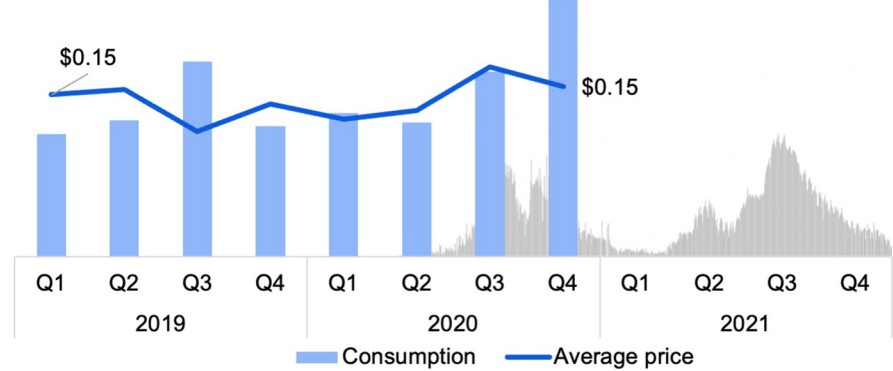

**Fig 5. Average cost of 1 day of treatment with NCD medicines and consumption levels, 2019–2021.** The data refer to the sum of consumption of medicines in DDDs used in treatment of diabetes (A10), in CVDs (B01, C01-C03, C07-C10), asthma and COPD (J01, R03), Cancer (hormone therapy for breast and prostate cancer–L02), epilepsy (N03), mental disorders (N05B, N06A) and weighted average prices.

In Georgia, 16 of the top 20 medicines with greatest overall expenditure had a price increase over 2019–21, with a median price change of +22%. In Azerbaijan, 9 of the 20 medicines with greatest overall expenditure had a price increase 2019–21, with a median price change of -1%.

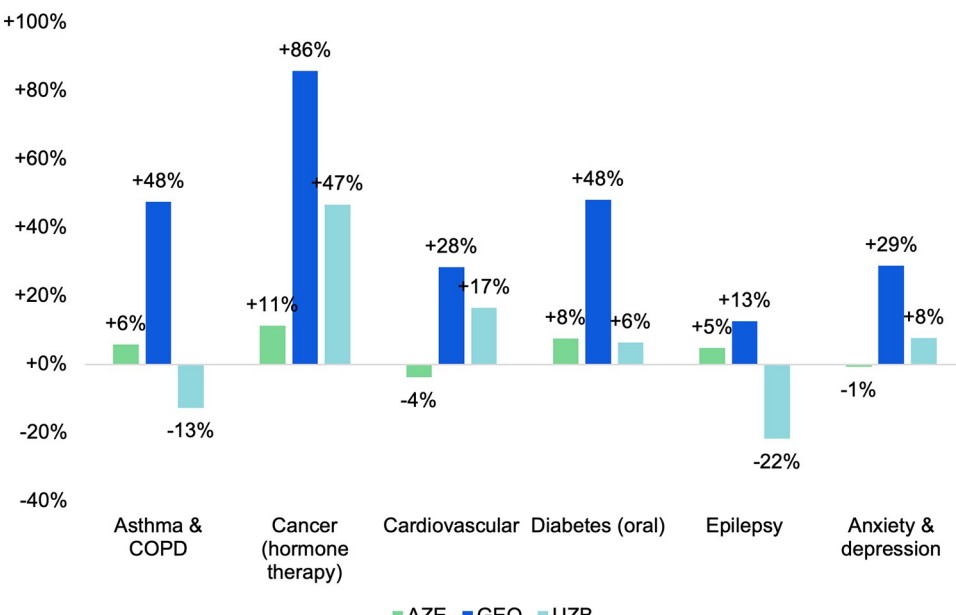

**Fig 6. Percentage change in average cost of 1 day of treatment for medicines used to treat NCDs since 2019.**
Azerbaijan and Georgia–difference between average price in 2021 compared to 2019; Uzbekistan–difference between average prince in 2020 compared to 2019.

In Uzbekistan, 9 of the 20 medicines with greatest overall expenditure had a price increase 2019–20, with a median price change of -4%. Several medicines appear in the list of top 20 medicines by overall expenditure for all three countries: two fixed-dose combinations including perindopril and amlodipine, bisoprolol, acetylsalicylic acid, clopidogrel, metformin, and amoxicillin with a beta-lactamase inhibitor (Table 2). Again, cardiovascular medicines dominated.

The top 20 medicines by absolute increase in overall expenditure are shown, for each country, in Table 3. The medicines with the highest cost per day of treatment are shown in Table 4.

## Discussion

This study analyzed trends in the consumption and prices of medicines for NCDs in Azerbaijan, Georgia, and Uzbekistan. All three are former Soviet countries, where the burden of ill health due to NCDs is above the global average, in terms of disability-adjusted life years (DALYs) lost [1]. Each country has taken different policy approaches to pharmaceutical pricing and reimbursement: Azerbaijan offers state health insurance covering services and certain inpatient medicines for the whole population, and regulates prices of all registered medicines throughout the supply chain using internal and external price referencing; Georgia offers coverage of a basic package of services for almost the whole population, and does not regulate medicine prices; Uzbekistan offers state health insurance covering primary and emergency health care for the whole population, as well as specialized services for vulnerable populations and certain inpatient medicines, for the whole population, and sets price limits for medicines as well as limits on wholesaler and pharmacy markups for all medicines [5, 9]. Many of the medicines covered in this analysis are on the WHO Essential Medicines List, and reliable, affordable access to these essential medicines is especially important in all health systems.

Georgia had the highest consumption for NCD medicines overall, and the highest consumption in every disease category except for asthma and COPD, where Azerbaijan had

**Table 2. Top 20 medicines by overall expenditures, 2019–2021.**

| Therapeutic group | Medicine | Total cost (USD) | Total cost change | Price change |
|---|---|---|---|---|
| **Azerbaijan** | | | | |
| CVDs | acetylsalicylic acid | 15,268,909 | +93% | -3% |
| CVDs | rosuvastatin | 12,866,724 | +43% | -3% |
| CVDs | perindopril, amlodipine and indapamide | 12,408,319 | +63% | +0% |
| Asthma & COPD | amoxicillin and beta-lactamase inhibitor | 10,991,022 | -11% | +10% |
| CVDs | bisoprolol | 9,646,087 | +33% | +12% |
| CVDs | lisinopril and amlodipine | 9,330,983 | +0% | -3% |
| CVDs | clopidogrel | 7,909,839 | +17% | -1% |
| CVDs | perindopril and amlodipine | 7,842,594 | +24% | -2% |
| CVDs | perindopril and diuretics | 7,822,924 | +47% | +6% |
| Mental disorders | escitalopram | 7,207,550 | +30% | -3% |
| Diabetes | metformin | 5,551,823 | +18% | -0% |
| Diabetes | metformin and sulfonylureas | 5,456,533 | +29% | +1% |
| CVDs | nebivolol | 5,314,321 | +30% | -3% |
| Asthma & COPD | ampicillin and beta-lactamase inhibitor | 5,012,627 | -24% | -3% |
| CVDs | atorvastatin | 4,876,804 | -0% | +1% |
| Asthma & COPD | montelukast | 4,792,231 | -0% | -6% |
| Asthma & COPD | salbutamol | 3,831,851 | -4% | +50% |
| Asthma & COPD | amoxicillin | 3,778,835 | -4% | +5% |
| Asthma & COPD | salmeterol and fluticasone | 3,668,788 | +16% | +5% |
| CVDs | captopril | 3,645,662 | +10% | -1% |
| **Georgia** | | | | |
| CVDs | perindopril, amlodipine and indapamide | 13,143,478 | +72% | +23% |
| CVDs | perindopril and amlodipine | 11,573,112 | +78% | -10% |
| CVDs | rivaroxaban | 11,451,011 | +157% | -2% |
| CVDs | rosuvastatin | 10,868,072 | +54% | +13% |
| CVDs | acetylsalicylic acid | 9,679,848 | +60% | +32% |
| Asthma & COPD | amoxicillin and beta-lactamase inhibitor | 9,579,396 | +0% | +35% |
| Asthma & COPD | azithromycin | 8,406,380 | +29% | +19% |
| CVDs | apixaban | 7,702,983 | -3% | -13% |
| CVDs | perindopril and diuretics | 6,195,181 | +44% | +39% |
| CVDs | enalapril and diuretics | 5,731,584 | +21% | +30% |
| CVDs | atorvastatin | 5,507,342 | +79% | -6% |
| Diabetes | metformin | 5,490,894 | +95% | +54% |
| CVDs | bisoprolol | 5,270,668 | +84% | +14% |
| CVDs | captopril | 4,886,614 | +55% | +30% |
| CVDs | clopidogrel | 4,872,346 | +38% | +16% |
| CVDs | losartan and diuretics | 4,780,550 | +147% | -10% |
| CVDs | metoprolol | 4,551,685 | +57% | +46% |
| CVDs | moxonidine | 4,218,278 | +39% | +14% |
| Mental disorders | escitalopram | 3,930,747 | +38% | +23% |
| Asthma & COPD | doxofylline | 3,715,926 | +19% | +39% |
| **Uzbekistan[a]** | | | | |
| Asthma & COPD | azithromycin | 31,456,487 | +297% | -26% |
| CVDs | bisoprolol | 14,684,417 | +21% | +35% |
| CVDs | acetylsalicylic acid | 10,305,466 | +18% | +4% |
| Diabetes | metformin | 9,326,299 | -9% | -4% |

*(Continued)*

**Table 2.** (Continued)

| Therapeutic group | Medicine | Total cost (USD) | Total cost change | Price change |
|---|---|---|---|---|
| CVDs | clopidogrel | 8,502,242 | +19% | +7% |
| Asthma & COPD | amoxicillin and beta-lactamase inhibitor | 8,414,468 | +20% | +15% |
| CVDs | losartan | 8,387,090 | +14% | +22% |
| CVDs | platelet aggregation inhibitors, combinations | 7,758,287 | +77% | -4% |
| CVDs | enalapril | 6,888,994 | -16% | +23% |
| CVDs | perindopril and amlodipine | 6,825,511 | +92% | -13% |
| Asthma & COPD | amoxicillin | 6,362,745 | -42% | -35% |
| Asthma & COPD | ampicillin | 6,124,258 | +57% | +61% |
| Diabetes | glimepiride | 5,704,390 | +10% | -4% |
| Epilepsy | pregabalin | 5,585,337 | -88% | -12% |
| CVDs | amlodipine | 5,470,569 | +37% | +21% |
| CVDs | rivaroxaban | 4,888,614 | +369% | -27% |
| Epilepsy | valproic acid | 4,615,759 | +4% | -7% |
| Asthma & COPD | salbutamol | 4,126,679 | +8% | -2% |
| Epilepsy | carbamazepine | 4,093,162 | -4% | -4% |
| CVDs | nebivolol | 4,075,903 | +37% | -4% |

a–time period for Uzbekistan is 2019–20.

slightly higher consumption (Fig 2). Overall consumption of NCD medicines in Georgia, expressed in DID, was 2 times as high as in Azerbaijan in 2021 and over 4 times as high as in Uzbekistan in 2020. These differences may be partially explained by the different NCD burden between the three countries. Georgia has the highest burden of NCDs: in terms of disability-adjusted life years lost, the burden in Georgia (33,346 DALYs per 100,000 population) is 59% above the global average (20,939 DALYs per 100,000 population), and markedly above the rates in Azerbaijan (23,602 DALYs per 100,000 population) and Uzbekistan (20,963 DALYs per 100,000 population). The burden of all disease categories included in this study, except epilepsy, is highest in Georgia, both in terms of DALYs and prevalence [1]. This is largely driven by the relatively older population in Georgia, where 16% of the population is over the age of 65 compared to 7% in Azerbaijan and 5% in Uzbekistan [17]; when data are adjusted for the age structure, Georgia's NCD rate is in fact below that in Azerbaijan or Uzbekistan [1]. It should be noted that Azerbaijan has state programs for preferential provision of medicines for certain groups of diseases, in particular, oncological, diabetic, thalassemia, and some types of disability. These medicines are distributed only through hospital pharmacies, and not community pharmacies, this may explain the lower consumption rates observed for medicines for these diseases in the community pharmacies in Azerbaijan.

We observed a high variability in consumption levels, with quarter-on-quarter decreases of 25% or more seen for a notable proportion of medicines in all therapeutic categories in all three countries (Fig 3), most pronounced in Uzbekistan overall, but with pronounced frequency for asthma and COPD medicines in Azerbaijan and Georgia. Given the conditions treated with the medicines are for the most part chronic conditions, this level variability in consumption suggests variable access to treatments. Addressing supply chain issues for NCD medicines, where they occur, should be a priority for health systems, as dependable access to treatment is key for these conditions.

The average prices of NCD medicines, weighted by consumption, increased by 26% in Georgia, but decreased by 3% in Azerbaijan and by 0.1% in Uzbekistan (Fig 4). Changes within

**Table 3. Top 20 medicines by increase in overall expenditures, 2019–2021.**

| Therapeutic group | Medicine | Overall expenditure increase (USD) | Change in price | Change in consumption |
|---|---|---|---|---|
| *Azerbaijan* | | | | |
| CVD | acetylsalicylic acid | +3,390,673 | -3% | +101% |
| CVD | perindopril, amlodipine and indapamide | +1,957,903 | +0% | +62% |
| CVD | rosuvastatin | +1,531,318 | -3% | +48% |
| CVD | perindopril and diuretics | +964,880 | +6% | +39% |
| CVD | bisoprolol | +906,298 | +12% | +19% |
| Mental disorders | escitalopram | +629,054 | -3% | +35% |
| CVD | perindopril and amlodipine | +544,003 | -2% | +26% |
| Asthma & COPD | budesonide | +531,938 | -5% | +95% |
| CVD | nebivolol | +464,835 | -3% | +34% |
| CVD | valsartan and sacubitril | +462,925 | -3% | +426% |
| CVD | olmesartan medoxomil and diuretics | +457,932 | +17% | +260% |
| Diabetes | metformin and sulfonylureas | +445,705 | +1% | +27% |
| Asthma & COPD | montelukast, combinations | +436,039 | -20% | +150% |
| CVD | clopidogrel | +424,101 | -1% | +18% |
| Diabetes | metformin and sitagliptin | +419,898 | -3% | +99% |
| CVD | candesartan and diuretics | +349,000 | -1% | +59% |
| Diabetes | metformin and vildagliptin | +346,046 | +0% | +69% |
| Epilepsy | pregabalin | +336,443 | -5% | +416% |
| Epilepsy | levetiracetam | +309,436 | +5% | +32% |
| Diabetes | metformin | +307,601 | -0% | +18% |
| *Georgia* | | | | |
| CVD | rivaroxaban | +3,818,974 | -2% | +152% |
| CVD | perindopril, amlodipine and indapamide | +2,301,837 | +23% | +39% |
| CVD | perindopril and amlodipine | +2,200,274 | -10% | +98% |
| CVD | acetylsalicylic acid | +1,575,005 | +32% | +22% |
| CVD | rosuvastatin | +1,546,459 | +13% | +36% |
| CVD | losartan and diuretics | +1,356,955 | -10% | +171% |
| Diabetes | metformin | +1,226,235 | +54% | +28% |
| CVD | atorvastatin | +1,061,019 | -6% | +90% |
| CVD | bisoprolol | +1,002,393 | +14% | +62% |
| Diabetes | gliclazide | +867,152 | +103% | +15% |
| CVD | perindopril and diuretics | +762,868 | +39% | +4% |
| Asthma & COPD | azithromycin | +743,476 | +19% | +8% |
| CVD | captopril | +692,624 | +30% | +20% |
| CVD | metoprolol | +685,524 | +46% | +7% |
| Asthma & COPD | piperacillin and beta-lactamase inhibitor | +674,471 | +5% | +291% |
| CVD | clopidogrel | +533,942 | +16% | +20% |
| CVD | omega-3-triglycerides incl. other esters and acids | +525,882 | +16% | +112% |
| Cancer | goserelin | +505,679 | +42% | +82% |
| CVD | moxonidine | +460,227 | +14% | +22% |
| Asthma & COPD | salmeterol and fluticasone | +448,355 | +24% | +53% |
| *Uzbekistan*[a] | | | | |
| Asthma & COPD | azithromycin | +18,805,365 | -26% | +510% |
| CVD | rivaroxaban | +3,169,366 | -27% | +598% |
| CVD | Platelet aggregation inhibitors, combinations | +2,160,291 | -4% | +76% |
| CVD | perindopril and amlodipine | +2,151,803 | -13% | +13% |

*(Continued)*

**Table 3.** (Continued)

| Therapeutic group | Medicine | Overall expenditure increase (USD) | Change in price | Change in consumption |
|---|---|---|---|---|
| CVD | losartan and amlodipine | +1,918,519 | +178% | -14% |
| CVD | bisoprolol | +1,377,495 | +35% | +15% |
| Asthma & COPD | ampicillin | +1,350,350 | +61% | -12% |
| CVD | spironolactone | +1,131,480 | -3% | +88% |
| CVD | apixaban | +1,094,176 | -5% | –* |
| Asthma & COPD | salmeterol and fluticasone | +953,014 | -1% | +120% |
| CVD | acetylsalicylic acid | +856,370 | +4% | +14% |
| CVD | amlodipine | +853,489 | +21% | -23% |
| CVD | valsartan and amlodipine | +771,023 | -13% | +88% |
| CVD | valsartan | +753,379 | -33% | +65% |
| Asthma & COPD | amoxicillin and beta-lactamase inhibitor | +750,344 | +15% | +9% |
| CVD | clopidogrel | +733,208 | +7% | +12% |
| Asthma & COPD | montelukast | +697,153 | +10% | +61% |
| CVD | valsartan and sacubitril | +658,199 | +192% | +11% |
| Diabetes | metformin and vildagliptin | +647,761 | -5% | +314% |
| CVD | nebivolol | +632,885 | -4% | +51% |

[a] –time period for Uzbekistan is 2019–20.

* –apixaban was introduced in Uzbekistan in 2020, so annual consumption change cannot be calculated.

individual therapeutic groups were more pronounced (Fig 5), with significant price increases in all disease groups in George (ranging from +13% for epilepsy medicines to +86% for hormone therapies for cancer), varied changes in Uzbekistan (ranging from -22% for epilepsy medicines to +47% for hormone therapies for cancer), and much less pronounced changes in Azerbaijan (ranging from -4% for CVD medicines to +11 for hormone therapies for cancer). In absolute terms, average costs per DDD were similar across the three countries for CVD, diabetes, epilepsy, and mental disorder medicines, while cancer medicines had markedly higher prices in Uzbekistan, and asthma and COPD medicines had markedly higher prices in Azerbaijan and Uzbekistan (Fig 6).

## Medicines driving cost increases

In all three countries, there is a high degree of overlap between the medicines with the greatest increases in expenditures 2019–21 and the medicines that had the highest overall expenditures in each country (Tables 2 and 3), suggesting that a relatively small group of medicines is responsible for a large part of year-to-year changes in total pharmaceutical expenditures in the community setting. In Azerbaijan, 11 of the 20 medicines with the highest overall expenditures were also in the top 20 medicines with greatest *rises* in expenditures. In Georgia, 15 of the 20 highest-expenditure medicines fell in this category. In Uzbekistan, 10 of the 20 highest-expenditure medicines fell in this category.

Overall expenditure on the NCD medicines included in this analysis increased from US$72 to 89 million in Azerbaijan (+24%), from US$61 to 89 million in Georgia (+45%), and from US$100 to 119 million in Uzbekistan (+18%).

To understand which medicines are driving cost increases, we look at the medicines with the greatest rise in absolute expenditures (Table 3). Most of these medicines are for cardiovascular disease, in all three countries. A key question to consider is whether cost increases are driven by price increases, increases in consumption, or a combination. For Azerbaijan, the

**Table 4. Top 20 medicines by average price per day of treatment 2019–2021.**

| Therapeutic group | Medicine | Price per DDD (USD) | Total cost (USD) | Price change |
|---|---|---|---|---|
| *Azerbaijan* | | | | |
| Cancer | fulvestrant | 10.54 | 7,494 | -22% |
| Diabetes | linagliptin | 7.39 | 1,432 | -32% |
| Cancer | triptorelin | 7.09 | 144,382 | +84% |
| CVD | omega-3-triglycerides incl. other esters and acids | 6.11 | 1,046,432 | +22% |
| Asthma & COPD | benzylpenicillin | 5.47 | 1,289,569 | -33% |
| CVD | nitroprusside | 4.29 | 5,247 | -2% |
| Asthma & COPD | ampicillin and beta-lactamase inhibitor | 3.74 | 5,012,627 | -3% |
| Asthma & COPD | theophylline, combinations with psycholeptics | 3.69 | 87,300 | +471% |
| Asthma & COPD | piperacillin and beta-lactamase inhibitor | 3.47 | 666,126 | -1% |
| CVD | apixaban | 3.34 | 7,522 | -14% |
| Asthma & COPD | josamycin | 2.99 | 494,973 | -6% |
| CVD | ticagrelor | 2.60 | 479 | -20% |
| Epilepsy | topiramate | 2.50 | 466,812 | +49% |
| CVD | rivaroxaban | 2.14 | 1,051,320 | -17% |
| CVD | nimodipine | 2.10 | 786,729 | -8% |
| Asthma & COPD | benzathine benzylpenicillin | 2.10 | 347,456 | +33% |
| Cancer | letrozole | 1.90 | 210,278 | -44% |
| Asthma & COPD | fluticasone | 1.88 | 1,120,978 | -48% |
| Cancer | goserelin | 1.82 | 123,341 | -1% |
| CVD | valsartan and sacubitril | 1.73 | 970,633 | -3% |
| *Georgia* | | | | |
| Cancer | abiraterone | 42.14 | 657,801 | +366% |
| CVD | evolocumab | 20.03 | 85 | +1% |
| Epilepsy | gabapentin | 15.49 | 207,280 | -79% |
| CVD | urapidil | 6.36 | 153,364 | +7% |
| Cancer | leuprorelin | 5.64 | 123,425 | +14% |
| CVD | apixaban | 5.64 | 7,702,983 | -13% |
| Diabetes | liraglutide | 3.63 | 382,004 | +2% |
| Repiratory | ampicillin and beta-lactamase inhibitor | 2.65 | 2,928,507 | +31% |
| Cancer | triptorelin | 2.44 | 129,583 | +11% |
| Asthma & COPD | josamycin | 2.33 | 137,813 | +41% |
| Asthma & COPD | indacaterol and glycopyrronium bromide | 2.28 | 34,639 | -23% |
| Asthma & COPD | piperacillin and beta-lactamase inhibitor | 2.26 | 1,390,360 | +5% |
| Cancer | fulvestrant | 2.19 | 1,151,719 | +16% |
| CVD | ticagrelor | 2.10 | 972,725 | +16% |
| Epilepsy | pregabalin | 1.73 | 385,916 | -19% |
| Epilepsy | topiramate | 1.57 | 224,647 | -46% |
| Diabetes | metformin and dapagliflozin | 1.47 | 361,452 | +39% |
| Asthma & COPD | ampicillin | 1.45 | 281,537 | +49% |
| Cancer | goserelin | 1.22 | 1,764,486 | +42% |
| Asthma & COPD | ipratropium bromide | 1.21 | 229,166 | +40% |
| *Uzbekistan*[a] | | | | |
| Asthma & COPD | montelukast, combinations | 27.53 | +112,418 | +1782% |
| Cancer | abiraterone | 25.14 | +116,906 | -3% |
| Cancer | buserelin | 14.78 | -169,764 | +6% |
| CVD | urapidil | 12.16 | -34,892 | -5% |

*(Continued)*

**Table 4.** (Continued)

| Therapeutic group | Medicine | Price per DDD (USD) | Total cost (USD) | Price change |
|---|---|---|---|---|
| CVD | valsartan and diuretics | 6.82 | +110,067 | +823% |
| Cancer | triptorelin | 6.25 | -19,602 | +16% |
| Asthma & COPD | ampicillin and beta-lactamase inhibitor | 6.24 | -1,150,699 | +0% |
| Diabetes | liraglutide | 5.42 | +58,881 | -8% |
| Cancer | fulvestrant | 4.83 | -88,496 | +4% |
| CVD | atorvastatin and amlodipine | 3.39 | -3,155 | +0% |
| CVD | dipyridamole | 3.31 | +611,026 | -24% |
| CVD | isosorbide dinitrate | 3.23 | +150,115 | -15% |
| CVD | apixaban | 3.11 | +1,094,176 | -5% |
| CVD | ticagrelor | 2.78 | +93,773 | -10% |
| Cancer | leuprorelin | 2.64 | +4,684 | +75% |
| Asthma & COPD | josamycin | 2.59 | +52,377 | -7% |
| Epilepsy | lamotrigine | 2.13 | +108,138 | -44% |
| CVD | rivaroxaban | 1.77 | +3,169,366 | -27% |
| CVD | perindopril and amlodipine | 1.73 | +2,151,803 | -13% |
| Epilepsy | topiramate | 1.63 | +69,828 | +17% |

[a]–time period for Uzbekistan is 2019–20.

picture in this regard is clear: increased costs are driven by increases in consumption, with the medicines that had the greatest cost increases all showing consumption increases (between 18% and 425%), while prices were less variable, ranging between a drop of 20% to an increase of 17% over 2019–21. In Georgia, consumption also increased significantly for all 'top 20' medicines, but this was accompanied in nearly all cases by significant price increases (with the exceptions of price reductions for the perindopril/amlodipine combination and atorvastatin). In Uzbekistan, consumption increased for 17 of the top 20 medicines by cost increase, with price changes split about equally between price reductions and price increases.

Overall, in all three countries, cost increases are driven by increased consumption. The trends in price changes suggest that Azerbaijan's price regulation policy has been effective, as key medicines (in terms of overall costs) are not showing significant price increases, and in many cases show slight price decreases, while usage is increasing notably (Table 2). In Georgia, where there was no price regulation during the study period, most of the top medicines in terms of overall costs showed significant price increases, suggesting that the lack of price regulation is a key factor in rising costs.

There is a risk of sampling bias in this type of analysis: While most of the 'top 20' medicines are older, standard primary care treatments with generally low absolute prices, this does not mean that high prices are not a significant access barrier. If a medicine is not prescribed or used due to its price being too high, it would not make it onto the 'top 20' lists and indeed would not even affect our analysis, as medicines with zero consumption are not included. In other words, certain unaffordable medicines may be 'invisible'.

## The effect of the COVID-19 pandemic

It is difficult to say with confidence whether the COVID-19 pandemic influenced medicines consumption in the three countries studied. Dips in consumption in 2020 Q2 can be seen in overall NCD consumption (Fig 1), as well in many of the disease group sub-analyses available in S2 and S3 Files. However, the dips in 2020 Q2 were small, and similar dips can in many

cases be seen occurring in 2019, before the emergence of the pandemic (Fig 3). This study did not undertake causal inference analysis to examine this question.

Granular analysis of individual medicines S3 File offers analysis at the level of individual medicines and chemical subgroups (ATC4), for example, bisoprolol and 'beta-blockers'.

In most disease categories, Georgia had the highest number of different products available in the health system for each medicine (S3 File). Considering that Georgia has the least regulation of the pharmaceutical market out of the three countries, this may reflect lower barriers to entry for competing manufacturers, compared to Azerbaijan and Uzbekistan.

Considering the key medicines in each disease group, we did not observe any precipitous drops in consumption for a particular quarter (S3 File), suggesting that supply has been steady. We can conclude that despite fluctuations in consumption, rising prices, and the risk of disruptions posed by the COVID-19 pandemic, access to essential medicines used to treat NCDs in outpatient settings has overall been robust over 2019–21. This does not rule out the possibility that there may have been shortages of certain medicines at certain times or in certain regions of the three countries, as our study does not use comprehensive data on all medications. Additionally, for some medicines that have low consumption at baseline (e.g., due to low prevalence of the treated disease), it may be harder to detect shortages based on consumption data.

### Health systems and pharmaceuticals

In terms of USD adjusted for purchasing power parity, health expenditures per capita were $606 in Azerbaijan, $970 in Georgia, and $418 in Uzbekistan, compared to an OECD average of $5,520, in 2019 [18]. When considered as a percentage of GDP, health expenditures per capita were 5% in Azerbaijan, 8% in Georgia, and 7% in Uzbekistan in 2020: closer to, but still below, the OECD average of 9.7% [8, 19]. In all three countries, per-capita spending on health rose around the period 2011–2016 and then declined, with 2019 expenditure returning to levels similar to expenditures in 2009 [20]. A high proportion of out-of-pocket health expenditures are on pharmaceuticals: 56% in Azerbaijan (2015 data) [6], 65% in Georgia (2018 data) [5] and up to 38% in Uzbekistan (proportion represents spending on all medical goods, 2018) [4]. These high proportions underline the importance of increasing government reimbursement of essential NCD medicines, in all three countries.

Per-capita spending on outpatient pharmaceuticals has increased by 16% in Azerbaijan (from $39 in 2019 to $46 in 2021), increased by 20% in Uzbekistan (from $31 in 2019 to $38 in 2020), and decreased by 14% in Georgia from (from $106 in 2019 to $91 in 2021) [8, 13].

Azerbaijan saw the lowest price increases among the main cost-driving medicines (Tables 2 and 3), and the lowest magnitude of change in the average cost per day of treatment for individual disease groups (Fig 5). This suggests that Azerbaijan's relatively extensive price regulation policy has controlled prices better than the other two countries' policy approaches. Medicine prices in Azerbaijan are set by the Tariff Council, and medicine price changes thus partially reflect the Council's activity. Where necessary, the Tariff Council implements a policy of price increases, especially during a pandemic, as one measure to lower the risk of shortages of certain medicines.

Earlier studies found that after introducing external price referencing (EPR) in 2015, Azerbaijan saw reduction in average medicine prices of 27% and 41% in the first and second years, respectively, and increases in consumption [5]. (Corresponding data are not available for Uzbekistan, where EPR is also used [5]). However, Azerbaijan had lower consumption than Georgia (Figs 1 and 2), and these differences cannot be fully explained by differences in disease burden: for example, while the prevalence rate of cancer is nearly identical in Azerbaijan

(8,944 per 100,000) and Georgia (9,167 per 100,000), 7 times less hormone therapies for cancer were consumed in Azerbaijan (Fig 2), with similar comparisons for all other disease groups, except for asthma and COPD medicines, where consumption was higher in Azerbaijan than in the other two countries.

In Azerbaijan, a national mandatory health insurance system, covering services and inpatient medicines, has been piloted since 2016, but only fully implemented in all regions in 2021 [21]. As part of the transition to a mandatory system, public health spending increased by 58% between 2019 and 2020 [22]. These changes are likely too recent to be reflected in our findings.

The health system in Georgia was the subject of an in-depth 2021 report by WHO Europe, which found that the affordability of medicines is a significant concern, and that out-of-pocket expenditure on medicines was the main contributor to catastrophic health expenditures. The high OOP spending on medicines in Georgia is "linked to the lack of price regulation, the frequent recommendation of brand-name medicines by physicians, and the limited availability of low-cost generic medicines in retail pharmacies", and noted that "[m]edicine prices are high compared to neighbouring countries and the cost-plus margin for pharmacies (frequently more than 100%) significantly exceeds margins established in EU countries" [23, 24]. Apart from the high level of OOP health expenditures in Georgia, there are other signs of limited access to medicines. For example, in 2019, 58% of patients received systemic therapy for cancer, compared to estimates that systemic therapy is needed for 72% of patients [25].

The Georgian Universal Health Care Programme (UHCP), introduced in 2013, covers only a limited basket of medicines (fewer than 100), covers only 50% of the costs of those medicines, and coverage of costs is limited to 200 GEL (about 75 USD) per year. UHCP budget records suggest that it provided very little support for outpatient medicine costs, with expenditures equivalent to partial cost assistance for 470 patients, at most. In February 2020, the annual budget cap was removed, and the administrative procedure for accessing support for costs of medicines was simplified with digital solutions, making access easier [5, 23]. Additionally, the range of medicines covered is being expanded, for example, in oncology [26].

Consumption in Georgia has risen markedly, implying an increase in access to treatments, perhaps reflecting the effects of a more generous coverage policy and streamlined administrative procedure. At the same time, pharmaceutical prices are rising rapidly (Figs 3 and 4 and Tables 2–4). Georgia's lack of price regulation means that access relies on healthy competition between manufacturers. In our analysis, prices in Georgia increased more than in Azerbaijan and Uzbekistan, where price regulation is used, suggesting that the free pricing policy is not successfully controlling pharmaceutical expenditures.

Uzbekistan provides universal coverage for all citizens, for a basic benefits package, which covers "socially significant and hazardous conditions", including infectious diseases such as TB and HIV, and certain noncommunicable diseases such as cancer. In practice, outpatient pharmaceuticals are excluded from coverage for significant parts of the population. Uzbekistan's level of out-of-pocket health expenditure (60%) is far above the average of LMICs in the region (51%) [4]. In 2018, 18% of households in Uzbekistan experienced catastrophic health expenditures [4]. Uzbekistan sets price limits for all medicines, using external price referencing [9]. Additionally, Uzbekistan regulates wholesaler and pharmacy markups for all medicines, with limits of 15% and 20% markups, respectively. Outpatient medicines are covered if they are on the list of 'socially important' medicines for 13 defined diseases [5]. Uzbekistan has started piloting of a reimbursement program in outpatient settings in 2021 [4, 5].

Uzbekistan showed particularly low consumption and high prices for hormone therapies for cancer treatment (Figs 2 and 6). While the burden of cancer is lower in Uzbekistan than in Azerbaijan or Georgia, the difference in consumption is far greater and differences in burden would not be enough to explain the disparity: The cancer prevalence rate in Uzbekistan was

12% lower than in Azerbaijan and 14% lower than in Georgia [1], while consumption of hormone therapies for cancer was 91% lower than in Azerbaijan and, remarkably, 99% lower than in Georgia (Fig 2). Disparities of this nature–where differences in disease burden are far smaller than differences in consumption–were seen for all medicine groups in this analysis, with Uzbekistan having the lowest consumption in all cases. This suggests that is significant undertreatment of NCDs in Uzbekistan, either due to a lack of access to diagnosis and care, barriers in accessing prescribed medicines, or a combination.

A 2014 study tracked household survey data from 2001 to 2010, to compare the proportion of the population forgoing medicines due to cost in eight former Soviet EECA countries–Armenia, Belarus, Georgia, Kazakhstan, Kyrgyzstan, Moldova, Russia, and Ukraine–finding that access to medicines had improved in most of the countries in this period, while there had been little improvement in Armenia and Georgia. The greatest improvements were seen in countries that implemented price controls, expanded benefits packages, and rational prescribing [27].

## Limitations

The study is limited to the outpatient setting and to community pharmacies. We did not include medicines used in hospital or dispensed from hospital pharmacies.

The SMD pharmaceutical database contains information on pharmaceutical products purchased in 2019─2021 at retail pharmacies, and therefore does not capture trends in prices and consumption for inpatient medicines. The list of medicines for which data were available is similar but not identical across the three countries compared, introducing the possibility of sampling bias affecting the cross-country comparisons.

Data on consumed pharmaceutical products in Uzbekistan were limited to the 2019–20 period.

We did not attempt causal inference analysis to assess the effects of the COVID-19 pandemic on pricing and consumption, or to assess the effects of individual national policies, such as inclusion of certain medicines on reimbursement lists.

## Conclusions

The highest consumption of NCD medicines in the outpatient setting was seen in Georgia, implying broader access to treatment than in Azerbaijan and Uzbekistan. However, Georgia also saw marked price increases, while Uzbekistan saw lesser price increases, and prices showed little changes in Azerbaijan.

Azerbaijan's policy combining external price referencing and limited and mark-ups in the retail network seems to have limited price increases, while consumption levels increased. However, consumption in Azerbaijan was lower than in Georgia, which cannot be explained by differences in NCD burden alone, implying inferior access to medicines.

In Uzbekistan, consumption was the lowest of the three countries, implying significant challenges in access to medicines.

Outpatient pharmaceutical expenditures are rapidly rising in all three countries. In Azerbaijan and Uzbekistan, increases in outpatient pharmaceutical expenditures were primarily driven by increases in consumption, rather than increases in price. In Georgia, where there was no price regulation, widespread price increases and increases in consumption both contribute to increasing pharmaceutical expenditures.

All three countries are in the midst of changes in their national health insurance systems, which aim to expand coverage. Comparing these three countries, each with a different combination of national policies, can help policymakers understand how access to NCD medicines

can be strengthened. There is a need for increased reimbursement of essential medicines for NCDs in all three countries, as a key component of improving NCD management.

## Supporting information

**S1 File. Appendix 1.**
(DOCX)

**S2 File. Appendix 2.**
(DOCX)

**S3 File. Appendix 3.**
(DOCX)

## Author Contributions

**Conceptualization:** Ninell Kadyrova, Dzintars Gotham, Stanislav Kniazkov.

**Formal analysis:** Ninell Kadyrova, Dzintars Gotham, Polad Hajibalayev, Zohid Ermatov, Natasha Azzopardi Muscat.

**Funding acquisition:** Stanislav Kniazkov.

**Investigation:** Dzintars Gotham.

**Methodology:** Ninell Kadyrova, Dzintars Gotham, Stanislav Kniazkov.

**Project administration:** Dzintars Gotham, Stanislav Kniazkov.

**Supervision:** Stanislav Kniazkov.

**Visualization:** Ninell Kadyrova, Dzintars Gotham.

**Writing – original draft:** Dzintars Gotham.

**Writing – review & editing:** Ninell Kadyrova, Dzintars Gotham, Stanislav Kniazkov, Elsever Aghayev, Polad Hajibalayev, Zohid Ermatov, Natasha Azzopardi Muscat.

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
