## [Decision Letter · Decision Letter 0]

7 Nov 2023

Trends in cost and consumption of essential medicines for non-communicable diseases in Azerbaijan, Georgia, and Uzbekistan, from 2019 to 2021

PONE-D-23-32458

Dear Dr. Kniazkov,

We’re pleased to inform you that your manuscript has been judged scientifically suitable for publication and will be formally accepted for publication once it meets all outstanding technical requirements.

Kind regards,

Arianit Jakupi, PhD

Academic Editor

PLOS ONE

   "This study was funded by WHO Europe."

Please respond by return e-mail so that we can amend your financial disclosure and competing interests on your behalf.

4. Thank you for stating the following in the Competing Interests/Financial Disclosure* (delete as necessary) section: 

   "NK has received funding from the World Health Organization, Health Transform Forum/GIZ, Childhood without diabetes/World Diabetes Foundation and declares no competing interests.

DG has received funding from the World Health Organization, Global Justice Now, Medicines Patent Pool, Médecins Sans Frontières, STOPAIDS, Treatment Action Group, and the World Intellectual Property Organization, and declares no competing interests.

SK is a salaried employee of WHO Europe and declares no conflict of interest.

EA receives funding from the Azerbaijan State Advanced Training Institute for Doctors named after A.Aliyev and declares no conflicts of interest.

PH is a PhD researcher of Pharmaceutical Faculty of the Azerbaijan State Advanced Training Institute for Doctors named after A. A. Aliyeva and declares no conflicts of interest.

ZE receives money from the State Medical Insurance Fund of Uzbekistan in the form of a salary and declares no conflicts of interest.

NAM is a salaried employee of WHO Europe and declares no conflict of interest." 

We note that one or more of the authors are employed by a commercial company: World Health Organization

a Please provide an amended Funding Statement declaring this commercial affiliation, as well as a statement regarding the Role of Funders in your study. If the funding organization did not play a role in the study design, data collection and analysis, decision to publish, or preparation of the manuscript and only provided financial support in the form of authors' salaries and/or research materials, please review your statements relating to the author contributions, and ensure you have specifically and accurately indicated the role(s) that these authors had in your study. You can update author roles in the Author Contributions section of the online submission form.

Please respond by return email with an updated Funding Statement and Competing Interests Statement and we will change the online submission form on your behalf.

Additional Editor Comments (optional):

Reviewers' comments:

Reviewer's Responses to Questions

**Comments to the Author**

1. Is the manuscript technically sound, and do the data support the conclusions?

Reviewer #1: Yes

Reviewer #2: Yes

2. Has the statistical analysis been performed appropriately and rigorously? 

Reviewer #1: Yes

Reviewer #2: Yes

3. Have the authors made all data underlying the findings in their manuscript fully available?

Reviewer #1: Yes

Reviewer #2: Yes

4. Is the manuscript presented in an intelligible fashion and written in standard English?

Reviewer #1: Yes

Reviewer #2: Yes

5. Review Comments to the Author

Reviewer #1: I have had the opportunity to thoroughly review manuscript titled "Trends in cost and consumption of essential medicines for non-communicable diseases in Azerbaijan, Georgia, and Uzbekistan, from 2019 to 2021." In the following paragraphs, I provide an academic assessment of this work, evaluating its technical soundness, data analysis, presentation, and language quality.

Technical Soundness: The manuscript demonstrates a high degree of technical soundness. The research methodology appears well planned and meticulously executed. The experimental design, data collection, and analysis are all appropriately documented. The study's objectives are clear, and the research questions are logically addressed. The use of appropriate methods and techniques in data collection and analysis enhances the manuscript's credibility.

Data and Conclusions: The data presented in the manuscript are comprehensive and well organized. It is evident that the data supports the conclusions drawn. The results align with the research objectives, and the conclusions are well founded. The logical flow of the manuscript aids in understanding the significance of the findings. This alignment between the data and the conclusions significantly strengthens the manuscript's overall quality.

Statistical Analysis: The statistical analysis in the manuscript is both appropriate and rigorous. The methods utilized for data analysis are well described, and the statistical tests chosen are relevant to the research questions. Moreover, the presentation of results in tables, figures, and the accompanying discussion contributes to the overall clarity of the statistical analysis.

Presentation and Clarity: The manuscript is well structured and presented in an intelligible fashion. The logical flow from introduction to methods, results, and discussion aids readers in following the research narrative. The figures and tables are appropriately placed and add value to the text, helping to visualize the data and results.

Language and Style: The manuscript is written in Standard English, demonstrating clarity and conciseness in its language and style. The use of appropriate scientific terminology and references contributes to the professionalism of the work. The absence of grammatical and typographical errors further enhances the manuscript's quality.

In conclusion, manuscript is of high quality, both in terms of its technical content and presentation. The study is methodologically sound, the data analysis is robust, and the conclusions are well supported by the data. The manuscript is well organized, written in Standard English, and free from language issues.

Reviewer #2: Shortcomings of the study was highlighted by the author that study is done by using only the form community pharmacies. Since the three countries have different health insurance systems direct compresence is hard without including data from Hospital's and Hospital's pharmacies.

Even with these shortcomings article achieve desired results.

6. PLOS authors have the option to publish the peer review history of their article (what does this mean?). If published, this will include your full peer review and any attached files.

Reviewer #1: No

Reviewer #2: No

---

## [Editor Report · Acceptance letter]

28 Nov 2023

PONE-D-23-32458 

Trends in cost and consumption of essential medicines for non-communicable diseases in Azerbaijan, Georgia, and Uzbekistan, from 2019 to 2021 

Dear Dr. Kniazkov:

I'm pleased to inform you that your manuscript has been deemed suitable for publication in PLOS ONE. Congratulations! Your manuscript is now with our production department. 

Kind regards, 

on behalf of

Dr Arianit Jakupi 

Academic Editor

PLOS ONE